# Mechanistic Insights in NeuroD Potentiation of Mineralocorticoid Receptor Signaling

**DOI:** 10.3390/ijms20071575

**Published:** 2019-03-29

**Authors:** Lisa T. C. M. van Weert, Jacobus C. Buurstede, Hetty C. M. Sips, Isabel M. Mol, Tanvi Puri, Ruth Damsteegt, Benno Roozendaal, R. Angela Sarabdjitsingh, Onno C. Meijer

**Affiliations:** 1Department of Medicine, Division of Endocrinology, Leiden University Medical Center, 2333 ZA Leiden, The Netherlands; L.T.C.M.van_Weert@lumc.nl (L.T.C.M.v.W.); J.C.Buurstede@lumc.nl (J.C.B.); H.C.M.Sips@lumc.nl (H.C.M.S.); I.M.Mol@lumc.nl (I.M.M.); tanvi-04@hotmail.com (T.P.); 2Department of Cognitive Neuroscience, Radboudumc, 6525 GA Nijmegen, The Netherlands; Benno.Roozendaal@radboudumc.nl; 3Donders Institute for Brain, Cognition and Behaviour, Radboud University, 6525 EN Nijmegen, The Netherlands; 4Department of Translational Neuroscience, UMC Utrecht Brain Center, University Medical Center Utrecht, 3584 CG Utrecht, The Netherlands; R.Damsteegt@umcutrecht.nl (R.D.); r.a.sarabdjitsingh@gmail.com (R.A.S.)

**Keywords:** basic-helix-loop-helix, brain, coactivator, glucocorticoids, hippocampus, mineralocorticoid receptor knockout, stress, transcription biology

## Abstract

Mineralocorticoid receptor (MR)-mediated signaling in the brain has been suggested as a protective factor in the development of psychopathology, in particular mood disorders. We recently identified genomic loci at which either MR or the closely related glucocorticoid receptor (GR) binds selectively, and found members of the NeuroD transcription factor family to be specifically associated with MR-bound DNA in the rat hippocampus. We show here using forebrain-specific MR knockout mice that GR binding to MR/GR joint target loci is not affected in any major way in the absence of MR. Neurod2 binding was also independent of MR binding. Moreover, functional comparison with MyoD family members indicates that it is the chromatin remodeling aspect of NeuroD, rather than its direct stimulation of transcription, that is responsible for potentiation of MR-mediated transcription. These findings suggest that NeuroD acts in a permissive way to enhance MR-mediated transcription, and they argue against competition for DNA binding as a mechanism of MR- over GR-specific binding.

## 1. Introduction

The mineralocorticoid receptor (MR) regulates stress coping and has gained significant attention in the field of psychopathology. In general higher brain MR expression levels or MR activity parallel improved cognition and reduced anxiety [1]. An MR gain-of-function variant is associated with optimism and provides a decreased risk for depression in females [2]. One single nucleotide polymorphism (SNP) that is part of this haplotype affected the cortisol-awakening response only in those subjects using antidepressants [3]. Furthermore, administration of an MR agonist as a supplement to antidepressant therapy led to faster treatment response [4], and MR activation alone could improve cognitive function in young depressed patients [5]. In contrast, chronic stimulation of the highly related glucocorticoid receptor (GR) predisposes to stress-related disorders [6], and GR antagonism seems of benefit in psychotic depression [7]. A study combining standard dexamethasone (GR activation) for leukemia treatment with add-on cortisol (concurrent MR activation), shows that MR activity is important for neuronal processes such as sleep cycle and mood regulation [8]. Therefore, it is of great relevance to characterize and enable selective modulation of MR-mediated effects, serving a potential antidepressant approach.

Being part of the nuclear receptor family, MR and GR function as ligand-activated transcription factors, binding the glucocorticoid response element (GRE) at the DNA to mediate transcriptional changes. Even though the two receptors share their ligand cortisol/corticosterone (albeit with a different affinity) and recognize the same motif, receptor-specific binding loci exist as demonstrated in the rat hippocampus [9]. This suggests that other factors might be necessary to guide MR/GR-specific binding and subsequent transcriptional effects. Indeed, we found that binding sites for NeuroD factors were present selectively near MR-bound loci, and confirmed Neurod2 binding near MR-bound but not GR-bound GREs [9]. Furthermore NeuroD factors were able to potentiate glucocorticoid-mediated signaling in an in vitro setting, although MR/GR specificity was not recapitulated in reporter assays [9].

NeuroD proteins belong to the basic-helix-loop-helix (bHLH) family of transcription factors, and regulate neuronal differentiation. Related MyoD factors are expressed in the muscle, where they induce myogenesis. The bHLH transcription factors bind to E-boxes, which have the sequence CANNTG [10]. Specificity is obtained via the middle two nucleotides, with CAGATG known to be a NeuroD-specific binding site, whereas CAGCTG is a shared site that is bound by both MyoD and NeuroD [11]. The previously found interaction between NeuroD and glucocorticoid signaling was based on the presence of the NeuroD-specific motif [9]. As the MyoD proteins are better understood in terms of functional domains [12], we also examined transcriptional modulation by bHLH factors at the MyoD/NeuroD shared motif to unravel the interaction between NeuroD and MR here.

The current study aimed to provide mechanistic insights in the NeuroD potentiation of MR signaling, and how MR over GR specificity is achieved. We selected the protein Neurod2 as a representative of the NeuroD family [9]. We first questioned whether GR binding would be affected by MR absence, and if Neurod2 binding would be dependent on MR presence. Therefore, we assessed GR and Neurod2 binding at previously identified MR targets [9] in the hippocampus of forebrain-specific MR knockout mice (fbMRKO). Subsequently using various E-box binders in a reporter assay, we further explored the mechanism by which NeuroD can enhance glucocorticoid signaling. Our data show that at MR target loci both GR and Neurod2 binding seem independent of MR binding, and it is likely the chromatin remodeling effect of NeuroD is responsible for the transcriptional potentiation.

## 2. Results

### 2.1. DNA Binding Assessed by Chromatin Immunoprecipitation

In order to define the mechanism behind the NeuroD potentiation of glucocorticoid signaling in more detail, we first tested whether MR binding to its hippocampal DNA targets affects local GR and Neurod2 binding. Although family members Neurod1, Neurod2 and Neurod6 are all expressed in the adult mouse hippocampus and are able to bind the same NeuroD binding site [9], we focus here on Neurod2. GR and Neurod2 occupancy of MR-binding loci was measured by chromatin immunoprecipitation coupled with quantitative polymerase chain reaction (ChIP-qPCR) on hippocampus of wild-type (WT) and forebrain-specific mineralocorticoid receptor knockout (fbMRKO) mice. The fbMRKO mice show ablated hippocampal MR mRNA levels [13], which is accompanied by efficient knockdown of MR protein (Bonapersona et al., in preparation). Plasma corticosterone of all animals was over 140 ng/mL, ensuring ligand occupancy of both MR and GR [14]. No difference in corticosterone plasma levels was observed between the two genotypes, with an average of 363 ± 30 ng/mL for WT mice and 313 ± 44 ng/mL for fbMRKO mice (Figure 1A).

#### 2.1.1. MR Effect on GR Binding

We aimed to investigate if the joint binding of MR and NeuroD on the DNA is related to competition for GR binding at the same locus. GR binding was confirmed in WT mice for classical glucocorticoid target genes *Fkbp5* and *Per1* (Figure 1B), which are occupied by both MR and GR [15]. Other MR-GR overlapping loci near the *Klf9* [16] and *Kif1c* [9] genes showed evident GR binding. Previously identified MR-specific target *Rilpl1* [9] showed low GR signal, to the same extent as MR-GR overlapping target *Zfp219* [9]. GR binding levels were similar in the fbMRKO mice for most of the genes measured, suggesting that GR binding is not dependent on MR binding at these target loci. Only the GR binding at *Per1* was slightly enhanced in MR absence (*p* = 0.00055), which might point to a compensatory mechanism at this specific binding site. However, in general GR binding does not seem to compensate for the lack of MR binding in fbMRKO mice.

#### 2.1.2. MR Effect on Neurod2 Binding

Next, we addressed the question of whether the association between MR and NeuroD factors that we observed previously implies that Neurod2 binding at these loci depends on the presence of MR. We measured Neurod2 binding at the same loci as for GR binding. No Neurod2 binding motif was detected in the ChIP-identified MR-GR overlapping binding sequences near *Fkbp5*, *Klf9* and *Per1*. For the *Kif1c* and *Zfp219* associated MR-GR overlapping loci a directed motif search [9] did reveal a Neurod2 binding motif. Neurod2 binding was indeed observed for *Kif1c*, *Zfp219* and to a lesser extent in *Klf9*, and for MR-specific *Rilpl1* as observed before [9] (Figure 1C). Those genes with relatively low GR binding showed higher Neurod2 binding and vice versa, supporting the earlier finding that Neurod2 seems to interact preferentially with MR [9]. The fbMRKO mice demonstrated unchanged Neurod2 binding levels, indicating the presence of MR is not crucial for Neurod2 binding. For *Kif1c* there might be an interaction, as the Neurod2 signal seems to be lower in fbMRKO compared to WT animals, but this difference does not statistically hold after multiple comparison correction (*p* = 0.23). Overall, these data show that Neurod2 binding to MR-associated loci is independent of MR binding.

### 2.2. Structure–Function Relationship

We continued unraveling the mechanism behind the NeuroD potentiation of glucocorticoid signaling by exploring which coactivation property of the NeuroD protein is responsible for the transcriptional potentiating effects. While the structure–function relationship of the NeuroD family is not known in detail, much more is known about the related bHLH family of MyoD proteins [12]. We therefore used the myogenic regulatory factors MyoD and Myf5 as tools to study the effect of bHLH factors in the potentiation of glucocorticoid signaling. Where MyoD can induce both histone acetylation at H4 (chromatin remodeling) and in addition recruit RNA polymerase II (direct activation mediated by the transcriptional activation domain), Myf5 is only able to induce H4 acetylation as a manner to enhance transcription [12]. NeuroD family members have been shown to affect both chromatin accessibility and direct transcriptional activation [11,17], although these functions have not been assigned to a specific part of the protein. Comparing the myogenic variants will enable us to dissect the process important for the potentiation of glucocorticoid signaling.

#### 2.2.1. Transcriptional Potentiation by MyoD

We started by exploring whether MyoD is able to show a similar coactivation effect for MR/GR-mediated signaling as Neurod2 did in our reporter assay. Despite the in vivo binding selectivity of Neurod2 with MR (and not GR), Neurod2 exhibits coactivation of MR but also GR transcriptional activity in vitro [9]. MyoD and NeuroD have both unique and common response elements [11]. Our original reporter construct that is based on in vivo MR ChIP-sequencing binding sites [9], harbors the NeuroD-specific CAGATG along a GRE. In a first experiment we tested the effect of Neurod2, MyoD and a chimeric MyoD protein with its bHLH domain substituted by that of Neurod2 (MyoD(ND2bHLH)) in the concentrations of 1–3–10 ng/well (Figure 2). Both a cofactor (*F*_2,24_ = 356.3 for MR; *F*_2,24_ = 708.3 for GR, both *p* < 0.000001) and concentration (*F*_3,24_ = 247.6 for MR; *F*_3,24_ = 489.0 for GR, both *p* < 0.000001) effect, plus an interaction (*F*_6,24_ = 71.0 for MR; *F*_6,24_ = 159.2 for GR, both *p* < 0.000001) were observed.

We confirmed Neurod2 could potentiate glucocorticoid signaling for both MR and GR (Figure 2A,B). The observed Neurod2 effect was receptor-mediated, as in absence or with lower amounts of nuclear receptor expression vector Neurod2 did not enhance the glucocorticoid-dependent transcriptional increase (Appendix A). We showed that also MyoD can potentiate MR- and GR-mediated transcriptional activity, once brought to the DNA. Coactivation by MyoD itself is minimal with a slightly higher fold induction in the upper tested dose compared to control cells without cofactor (*p* = 0.0062 for MR; *p* = 0.0019 for GR), but can be enhanced to an extent similar to Neurod2 by swopping the MyoD DNA-binding domain (DBD) with that of Neurod2 as demonstrated using the MyoD(ND2bHLH) chimera (Figure 2). In its highest tested dose the chimera could even potentiate glucocorticoid signaling to a superior extent. Of note, the chimera showed a clear dose-dependent increase in potentiation over the concentration range tested. These findings indicate the Neurod2 DBD is required for coactivation, and the DNA sequence rather than the bHLH protein function drives specificity.

#### 2.2.2. Activation Domain Not Crucial for Potentiation

Finally we tested several bHLH factors for their coactivation ability in our reporter assay to examine the contribution of different protein domains. In order to have a fair comparison of all variants, we ensured a similar binding affinity of NeuroD and MyoD by further studying a reporter construct containing the shared CAGCTG motif [11]. At this reporter Neurod2 and MyoD could potentiate MR signaling to the same extent (Figure 3A), while for GR-mediated transcription the MyoD potentiation was somewhat lower than by Neurod2 (*p* = 0.000003, Figure 3B). MyoD lacking its activation domain (MyoDΔN) demonstrated a less strong potentiation of GR-mediated signaling compared to full length MyoD (*p* = 0.0012), as did family member Myf5 (*p* = 0.0035), but both MyoDΔN (*p* = 0.047) and Myf5 (*p* = 0.016) still showed a significantly higher transcriptional effect upon corticosterone treatment than the control condition without overexpression (Figure 3B).

The effect of the bHLH proteins on MR transactivation was more modest. Interestingly, the MyoDΔN and Myf5 coactivating potential for MR-mediated signaling was not different from Neurod2 and MyoD (Figure 3A). However, MyoDΔN did not reach significance in corticosterone induction compared to control cells (Figure 3A). Although potentiation of GR transcriptional activity by bHLH factors seems thus partly dependent on their activation domain, these data suggest that the coactivation of MR signaling by Neurod2 postulated to happen in vivo [9] is likely mediated via chromatin remodeling rather than direct transcriptional activation.

## 3. Discussion

This study further elucidates the mechanism behind NeuroD potentiation of brain MR signaling. First transcription factor DNA binding was assessed by ChIP-qPCR in mice lacking MR in (amongst other brain regions) their hippocampus. Both GR and Neurod2 binding were not altered in these fbMRKO mice compared to control mice, except for an enhanced GR signal at the *Per1* promoter in absence of MR. Subsequently bHLH factors of the NeuroD and MyoD families were used to study coactivator effects in an MR/GR-driven reporter assay. Those factors lacking (MyoDΔN) or with diminished (Myf5) activator function were able to potentiate the glucocorticoid-stimulated transcriptional activation as well as Neurod2 and MyoD in case of MR-dependent transcription, suggesting coactivation of MR signaling by Neurod2 does not require its activation domain.

### 3.1. Effects on DNA Binding

Because MR and GR can bind the same DNA sequences, GREs, the absence of MR might affect genomic binding by GR. Competition between MR and GR at a specific locus does not seem to play a major role, as there was no overall enhanced GR binding in the fbMRKO mice at the sites we examined, even though hippocampal GR expression is upregulated in these animals [13,18]. Only in the case of *Per1*, higher GR occupancy levels were observed at the promoter region in the absence of MR. At this locus it has been demonstrated that besides homodimerization, MR and GR can combine to form heterodimers [15]. However, we cannot distinguish between these two binding modes in our measurements. The increased GR binding could reflect a compensatory mechanism to maintain a required degree of *Per1* expression and is in agreement with the fact that basal *Per1* mRNA levels were not altered in fbMRKO mice [13]. Rather than competition, data on joint occupancy suggest there can be synergism between two transcription factors binding the same site, via a process called ‘assisted loading’. For concurrent stimulation of the GR and estrogen receptor (ER; where ER is altered to also recognize the GRE), GR activation could enhance ER binding at the same locus [19]. In the present study, GR binding is not significantly diminished when MR is lacking, suggesting such assisted loading is not applicable for MR-GR joint loci here. In our measurements of whole hippocampus we should acknowledge that we work under the assumption that all studied cells have (similar amounts of) MR and GR, but effects on DNA binding could be diluted as MR/GR expression is not homogeneous throughout the hippocampal regions and in the various cell types present [20]. Single cell analysis will offer a solution to study transcription biology in a cell-type specific manner [21]. Nevertheless, our data indicate that GR binding is predominantly independent from the presence of MR in the hippocampus.

In the same setting we studied if Neurod2 binding was affected by absence of MR. No differences in Neurod2 signal at the MR target loci were observed in MR deficient mice, which implies that NeuroD facilitates MR binding in a unidirectional manner. We cannot exclude the possibility that Neurod2 binding is affected by or dependent on changes in stress hormone levels, since this was not studied here. The presence of another collaborative transcription factor (nuclear factor-1) found near preaccessible GR-bound loci was independent of corticosterone treatment or exposure to restraint stress [22]. As discussed below, our reporter assay data suggest that the potentiation of MR signaling by NeuroD is likely mediated via chromatin accessibility.

### 3.2. Mechanism of Glucocorticoid Signaling Potentiation

Unfortunately, the NeuroD activation domain is not well documented/distinguished, but MyoD family members do have well described domains [12]. We first tested whether MyoD was able to potentiate glucocorticoid signaling at a reporter construct containing a GRE and NeuroD-specific E-box (CAGATG). When the MyoD DBD was adapted to that of Neurod2 in order to bind this motif efficiently, MyoD could coactivate glucocorticoid-mediated signaling to a similar (or even superior) extent as Neurod2. This is in line with findings by Fong et al., showing that MyoD could be redirected to NeuroD target sites through replacement of its bHLH domain by the analogue sequence of Neurod2 and, hereby, could activate part of the neuronal differentiation program [23]. The same group has demonstrated that NeuroD and MyoD can bind and drive transcription at the E-box that is specific for the other bHLH factor, but have a strong preference for their specific motifs [11,23]. This explains why unmodified MyoD showed a slight transcriptional potentiation on the NeuroD-specific binding site at its highest concentration tested. In concordance with the DBD being decisive in converting MyoD into a neurogenic factor [23], the specificity of the interaction between NeuroD/MyoD and MR/GR in our data is also determined by the ability of the factor to bind the DNA rather than a protein-specific functionality. Interactions between bHLH transcription factors and steroid receptors can be speculated to be generic but have cell/tissue-type dependent mechanisms. For instance, bHLH proteins DEC1/DEC2 (differentiated embryo chondrocyte) were found to corepress liver retinoid X receptors [24]. Likewise, E47 can modulate hepatic glucocorticoid action by promoting GR occupancy of metabolic target loci [25]. Of relevance in the testis, Pod-1 (also: transcription factor 21) could diminish transactivation by the androgen receptor [26].

For unbiased comparisons we proceeded our experiments with a reporter construct containing the shared E-box (CAGCTG), which is bound with similar affinity by both Neurod2 and MyoD [23]. Coregulators can modulate transcription by affecting chromatin accessibility and/or recruitment and stabilization of the transcriptional machinery [27]. To distinguish between these two modes, we made use of a truncated version of MyoD lacking its activation domain (responsible for direct recruitment), and the myogenic Myf5 that has a weak activation domain (and, therefore, relies mainly on its chromatin remodeling ability) compared to MyoD [12]. All MyoD variants were able to coactivate the GRE-driven reporter. Strikingly, while potentiation of GR signaling was partly dependent on the bHLH activation domain, coactivation of MR signaling was almost unaffected when using the factors with diminished direct transcriptional activation. Extrapolating these findings to the NeuroD family, the chromatin remodeling aspect of NeuroD thus seems sufficient for effective potentiation of MR-mediated signaling. This is in accordance with the pioneer function of family member Neurod1 demonstrated in a ChIP-sequencing experiment on developing neurons [17]. Of note, during neurogenesis occupancy of the Neurod2-specific motif was linked to gene expression effects, while the shared motif related mostly to chromatin modifications [11]. Despite the fact that transient systems might be considered to have an undefined chromatin context, it has been shown that exogenous plasmids do interact with endogenous histone proteins [28,29] and can serve as a proper model to study effects mediated via chromatin accessibility as observed here.

### 3.3. MR Selective Signaling and Future Implications

A number of issues have remained unaddressed. In the current study we have been looking at only a subset of Neurod2 sites, and mainly focused on targets bound by both MR and GR. It would be of interest to study genome-wide effects and observe if MR-specific sites become GR-bound in the absence of MR. We also have to point out that we have not assessed in vivo which NeuroD factor(s) is/are responsible for potentiation of MR signaling, as we only measured and detected Neurod2 binding at MR-bound sites [9]. The basis for MR over GR specificity in full chromatin is not known, but the fact that bHLH chromatin remodeling plays a more important role in case of MR-mediated reporter activation is in line with the fact that we could correlate MR and Neurod2 binding in vivo [9]. Besides, those MR target genes with relatively low GR signal had high Neurod2 binding in our current ChIP data. A study by Pooley et al. found that 17% of GR-bound loci contained a NeuroD binding site in their vicinity [22]. These are likely MR/GR joint sites comparable to those studied here, some of which do show an E-box and could be co-bound by Neurod2. MyoD family inhibitor domain-containing protein (MDFIC) has been found to bind the hinge region of unliganded GR, is capable of regulating GR phosphorylation and can by this means define the receptor transcriptome [30]. This interaction might play a role in the MR/GR binding selectivity near Neurod2-bound sites, as our earlier studies suggested that proteins in the nuclear receptor complex might account for the MR preference [9]. One promising approach to further elucidate the MR over GR specificity would be to have ChIP experiments followed-up by proteomics [31].

The question emerges what the NeuroD potentiation of MR signaling implicates for stress processing and stress-related disorders. Increased *Neurod2* expression levels were detected in the ventromedial prefrontal cortex of men with major depressive disorder compared to healthy control subjects [32]. In a mouse model of chronic social defeat paradigm, overexpression of Neurod2 in the ventral hippocampus reduces, while overexpression in the nucleus accumbens increases social interaction time [33]. Antidepressant agomelatine could normalize the rise in hippocampal *Neurod1* expression of mice that underwent chronic mild stress [34]. Furthermore, fish in touristic zones were shown to express higher levels of *Neurod1* and the MR gene *Nr3c2* relative to fish at control sites [35]. Together these observations strongly suggest a functional and context-dependent link between NeuroD and stress regulation. How this might depend on MR or influence MR function remains to be investigated. Further research is needed focusing on the in vivo specificity of the interaction between MR and NeuroD, and directionality in the highly adaptable stress system. MR activation is considered a promising strategy to promote stress resilience [1]. It would be of great interest to test if SNPs in the MR gene can affect NeuroD potentiation. In conclusion, we show that GR and Neurod2 binding at MR target loci is not dependent on MR presence and that Neurod2 potentiation of MR signaling is likely mediated via chromatin remodeling. We summarize the findings of this study in Figure 4. Future studies will have to point out how the interaction between Neurod2 and MR might be exploited to modulate MR-specific effects in the brain and affect associated behavior. 

## 4. Materials and Methods

### 4.1. Animals

Male homozygous forebrain-specific MR knockout (MR^flox/flox_Cre^, fbMRKO, *n* = 9) and littermate flox heterozygous control mice (MR^flox/wt_wt^, *n* = 10) [18] aged 10-19 weeks, were housed on a 12-h light/12-h dark reversed cycle (lights off at 8:00AM). Mice were group-housed with fbMRKOs and controls combined, and a total of four mice per cage. Each mouse was individually transferred to a novel cage 45 min before harvesting the tissue, in order to ensure GR binding for ChIP analysis. Mice were sacrificed by cervical dislocation around the time of their endogenous corticosterone peak, between 9:00AM–11:30AM. Genotypes were equally distributed over the sacrifice window to prevent an effect by time of the day. Trunk blood was collected, and hippocampal hemispheres were freshly dissected, snap-frozen in liquid nitrogen and stored at −80 °C for later analysis. The experiment was performed according to the European Commission Council Directive 2010/63/EU and the Dutch law on animal experiments and approved by the animal ethical committee from Utrecht University (authorization number 2014.I.08.057, approval date: 9 October 2014).

### 4.2. Plasma Corticosterone

Trunk blood was centrifuged for 10 min at 7000× *g*, after which plasma was transferred to new tubes and stored at −20 °C for later analysis. Corticosterone levels were determined using an Enzyme ImmunoAssay, according to the manufacturer’s instruction (Immunodiagnostic Systems, Boldon, UK).

### 4.3. ChIP-qPCR

To assess GR and Neurod2 binding at MR-bound loci, we performed ChIP-qPCR on hippocampal tissue as described previously [9]. Briefly, two fixated hippocampal hemispheres of the same animal were pooled and used for a single ChIP sample (500 µL) to measure GR binding (*n* = 4–5) with 6 µg of anti-GR antibody H-300 (sc-8992X, Santa Cruz, Dallas, TX, USA) or Neurod2 binding (*n* = 4) with 6 µg of anti-Neurod2 antibody (ab109406, Abcam, Cambridge, UK). Hippocampi were allocated for either GR or Neurod2 detection, with tissue from each group of co-housed mice divided over the two transcription factors. A ChIP using 6 µg of control IgG antibody (ab37415, Abcam) was taken along for background measurements, on a mixed hippocampal chromatin sample per genotype and transcription factor. This was followed by qPCR on undiluted Chelex-isolated (200 µL) ChIP samples, using the primers listed in Table 1.

### 4.4. Reporter Assays

For mechanistic insights into the role of NeuroD factors on MR/GR-driven promoter activity, we performed luciferase reporter as described previously [9]. In short, HEK293 cells were transfected using FuGENE (Promega, Leiden, The Netherlands) with luciferase construct (GRE-At, 30 ng/well), expression vector for either MR or GR (10 ng/well), with or without NeuroD/MyoD cofactor (10 ng/well), and Renilla (1 ng/well) for normalization. To exclude glucocorticoid effects from the medium we used charcoal-stripped fetal bovine serum (Sigma-Aldrich, Zwijndrecht, The Netherlands) during the experiments. After 24 h stimulation of the cells with 10^−7^ M corticosterone (Sigma) reporter protein levels were measured using the Dual Luciferase Reporter Assay System according to the manufacturer’s instruction (Promega).

### 4.5. Plasmids

Transcriptional activity was assessed at a GRE-driven promoter combined with either the NeuroD-specific (CAGATG) or the MyoD/NeuroD-shared (CAGCTG) motif. The GRE and NeuroD binding site-containing vector (GRE-At_GA) was constructed before (GRE-At-pGL4 [9]). For the generation of the GRE-At_GC luciferase construct, we exploited mutagenesis targeting the NeuroD binding site (GA > GC) using a QuikChange II Site-Directed Mutagenesis Kit (Agilent Technologies, Santa Clara, CA, USA). PAGE-purified mutagenic primers were: 5′-CTCGAGGATGGCAGCTGGAGCTAAGAACAGAA-3′ and 5′-TTCTGTTCTTAGCTCCAGCTGCCATCCTCGAG-3′. For MR and GR expression we used the 6RMR and 6RGR-based plasmids [36]. Expression vectors (all pCS2) for Neurod2, MyoD, a chimera of MyoD with the DNA-binding domain of Neurod2 (MyoD(ND2bHLH)), MyoD lacking the N-terminal domain (MyoDΔN) and Myf5 were kindly provided by Dr. Tapscott [12,23].

### 4.6. Statistics

On the ChIP data we ran unpaired *t*-tests with the Holm–Sidak multiple comparison correction. For the reporter assays we performed statistics on the fold induction by ligand (calculated for each corticosterone-treated sample as signal in the presence of hormone divided by the average signal from the same condition in absence of hormone). The first reporter experiment (different cofactors at various concentrations) was analyzed by two-way analysis of variance (ANOVA); the second reporter experiment (different cofactors) was analyzed by one-way ANOVA, both followed by Tukey’s post-hoc tests. All data are presented as mean ± standard error of the mean.

## Figures and Tables

**Figure 1 ijms-20-01575-f001:**
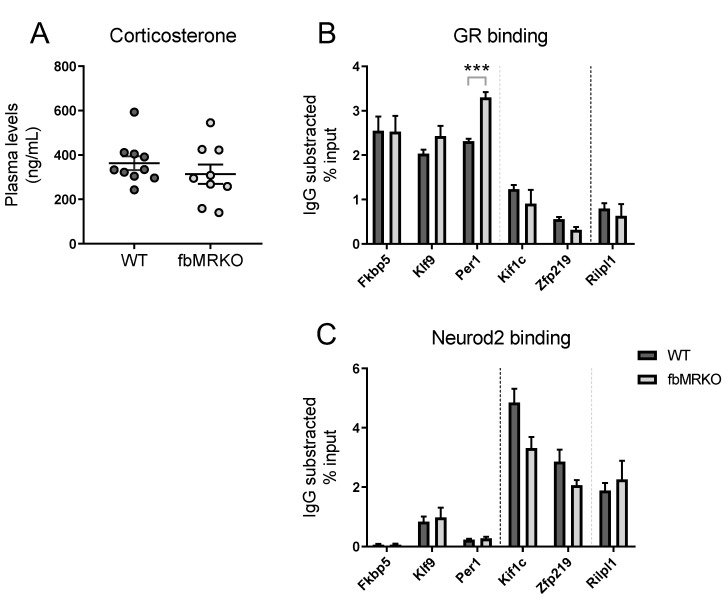
(**A**) Corticosterone levels of wild-type (WT) and forebrain-specific mineralocorticoid receptor (MR) knockout (fbMRKO) mice. In these mice chromatin immunoprecipitation coupled with quantitative polymerase chain reaction (ChIP-qPCR) measurements for (**B**) glucocorticoid receptor (GR) and (**C**) Neurod2 were performed. For each gene, the corresponding immunoglobulin G (IgG) background signal is subtracted from detected binding levels, expressed as the percentage of immunoprecipitated DNA. The binding sites near *Fkbp5*, *Klf9*, *Per1*, *Kif1c* and *Zfp219* are joint MR/GR loci, while *Rilpl1* has been identified as an MR-specific target [9] (separated by the right dotted line). Genes are further sorted based on the absence (*Fkbp5*, *Klf9*, *Per1*) or presence (*Kif1c*, *Zfp219*, *Rilpl1*) of a NeuroD binding sequence near the MR binding site (separated by the left dotted line). *** *p* < 0.001

**Figure 2 ijms-20-01575-f002:**
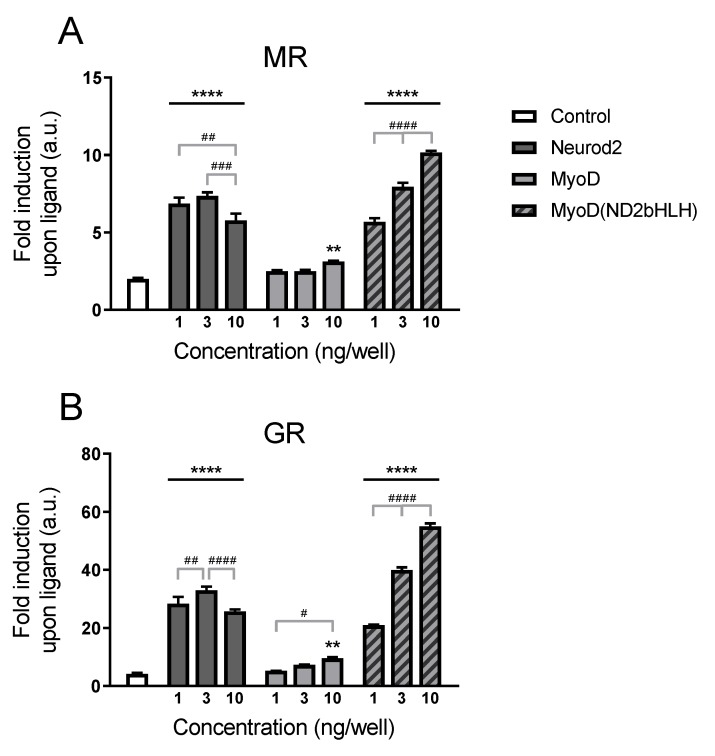
Specificity of NeuroD coactivation at the previously identified binding motif (CAGATG) for (**A)** MR and (**B)** GR. HEK293 cells were transfected with GRE-At_GA luciferase construct, MR or GR (10 ng/well), various amounts of Neurod2, MyoD or the MyoD/Neurod2 chimera (MyoD(ND2bHLH)) (1–3–10 ng/well), and stimulated with corticosterone (10^−7^ M). Data are presented as luciferase activity fold induction upon corticosterone treatment. a.u. = arbitrary unit; ** *p* < 0.01, **** *p* < 0.0001 compared to control condition; # *p* < 0.05, ## *p* < 0.01, ### *p* < 0.001, #### *p* < 0.0001 for within group comparisons

**Figure 3 ijms-20-01575-f003:**
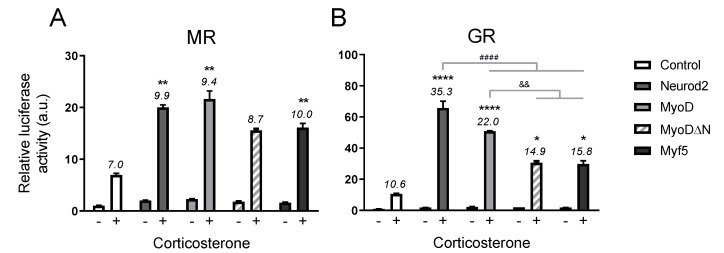
Modulation by NeuroD and MyoD variants at the shared binding motif (CAGCTG) for (**A**) MR- and (**B)** GR-mediated transcription. HEK293 cells were transfected with GRE-At_GC luciferase construct, MR or GR (10 ng/well), and Neurod2, MyoD, MyoDΔN or Myf5 (10 ng/well), and stimulated with corticosterone (10^−7^ M). Luciferase activity of nonstimulated control cells was normalized to 1. Numbers represent fold induction upon corticosterone treatment. a.u. = arbitrary unit; * *p* < 0.05, ** *p* < 0.01, **** *p* < 0.0001 compared to control condition; #### *p* < 0.0001 compared to Neurod2 condition; && *p* < 0.01 compared to MyoD condition

**Figure 4 ijms-20-01575-f004:**
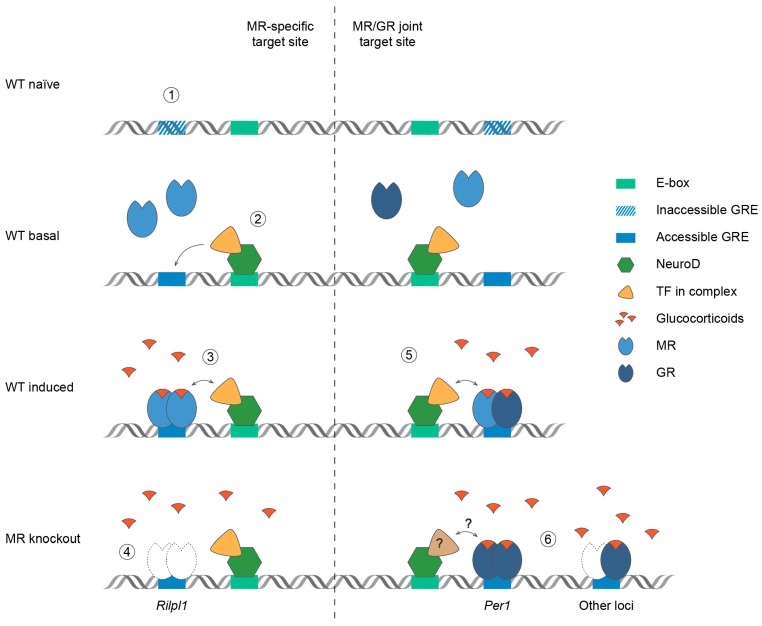
Summary of the interaction between hippocampal MR and NeuroD. Glucocorticoid response elements (GREs) previously inaccessible (**1**) could be rendered accessible by chromatin remodeling (one-way arrow) induced by NeuroD (**2**) binding at a nearby E-box (the NeuroD-specific sequence CAGATG). Upon ligand availability MR can bind an accessible GRE (**3**) in order to modulate transcriptional activity of its target genes. This interaction between NeuroD and MR (two-way arrow) is likely mediated via additional TF(s) in the transcriptional complex [9]. In forebrain MR knockout mice (**4**) GR is not compensating for the lack of MR binding at the MR-specific *Rilpl1* site. Also at several MR/GR joint target sites (**5**) NeuroD occupancy is observed in the vicinity. Of note, we cannot discriminate between the binding of homo- and heterodimers in the present study. In absence of MR (**6**) GR binding is increased at the *Per1* promoter, while for the other tested loci GR binding levels are unaltered. For sites that become GR-specific due to MR knockout, interactions with NeuroD remain to be explored, and other TF(s) might be involved (**?**). MR = mineralocorticoid receptor, GR = glucocorticoid receptor, GRE = glucocorticoid response element, TF = transcription factor, WT = wild type

**Table 1 ijms-20-01575-t001:** Primer sequences used for qPCR on mouse hippocampal ChIP samples. Primers target a mineralocorticoid receptor binding site near the listed gene.

Gene	Full Name	Forward & Reverse (5′ > 3′)	Product Length (bp)
*Fkbp5*	FK506 binding protein 5	TGCCAGCCACATTCAGAACA TCAAGTGAGTCTGGTCACTGC	122
*Kif1c*	Kinesin family member 1C	GCTGGGGTGTACACAGATGG TGACTAGCCAGAGCAGTATGTC	156
*Klf9*	Kruppel-like factor 9	ATCTAGGGCAGTTTGTTCAA GGCAGGTTCATCTGAGGACA	96
*Per1*	Period circadian clock 1	GGAGGCGCCAAGGCTGAGTG CGGCCAGCGCACTAGGGAAC	73
*Rilpl1*	Rab interacting lysosomal protein-like 1	CAGGCAGATGCCAGGCT CCCATGCCTGTTCCTCTAGT	106
*Zfp219*	Zinc finger protein 219	AGTCCATCACATTCTGTTGCTTTC TAGTCAGCTATGACCATGCAGT	131

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
