# Peer review of "Mechanistic Insights in NeuroD Potentiation of Mineralocorticoid Receptor Signaling"

_ijms, 2019, doi:10.3390/ijms20071575_

Round 1

Reviewer 1 Report

1.  How efficient is MR knockout in these animals? Have the authors checked this or are they relying on previously published data (Berger et al. PNAS 2006)?

2.  There are minor issues with the structure of the manuscript that could be improved. For instance, lines 5-51 and line 65 in the Introdution repeat the same information. The first two sections of Results (2.1 and 2.2) should be merged and Fig.1 should be reordered (the first result mentioned, corticosterone circulating levels, appear as panel C instead of panel A). Also, sections 2.4 and 2.5 should be merged, as it stands section 2.4 of Results is just the justification for performing the experiments shown in section 2.5.

3.  Perhaps it would be informative to include information about the mRNA levels of the genes tested by ChIP. For instance, the authors mention in the Discussion that Per1 levels are not altered in fbMRKO mice as an “unpublished observation”. Why not include it in Fig.1? Do the authors have data on the expression of the rest of the genes closest to the genomic loci tested?

4.  Line 157 and 158, “chimer” should read “chimera”

Author Response

Please find our reply to reviewer 1 comments in the uploaded pdf document.

Reviewer 2 Report

The current study aimed to provide mechanistic insights in the NeuroD potentiation of mineralocorticoid receptor signaling, and how mineralocorticoid receptor over glucocorticoid receptor specificity is achieved. Overall, I found the study very interesting and scientifically sound. The methods are very well explained and results as well. I have no great comments on the paper that deserves publication. I only suggest Authors to

1) Add a figure explaining how all variables may be correlated in the general psychopathology (only if Authors want, as I believe that may be useful for the reader)

2) I believe that the indroductory part on psychopathology would benefit from a table explaining more relevant studies that were reported in the first part of the introduction.

Author Response

Please find our reply to reviewer 2 comments in the uploaded pdf document.
